# Mendelian randomization analyses explore the relationship between cathepsins and lung cancer

Jialin Li[1], Mingbo Tang[1], Xinliang Gao[1], Suyan Tian [2✉] & Wei Liu [1✉]

Lung cancer, a major contributor to cancer-related fatalities worldwide, involves a complex pathogenesis. Cathepsins, lysosomal cysteine proteases, play roles in various physiological and pathological processes, including tumorigenesis. Observational studies have suggested an association between cathepsins and lung cancer. However, the causal link between the cathepsin family and lung cancer remains undetermined. This study employed Mendelian randomization analyses to investigate this causal association. The univariable Mendelian randomization analysis results indicate that elevated cathepsin H levels increase the overall risk of lung cancer, adenocarcinoma, and lung cancer among smokers. Conversely, reverse Mendelian randomization analyses suggest that squamous carcinoma may lead to increased cathepsin B levels. A multivariable analysis using nine cathepsins as covariates reveals that elevated cathepsin H levels lead to an increased overall risk of lung cancer, adenocarcinoma, and lung cancer in smokers. In conclusion, cathepsin H may serve as a marker for lung cancer, potentially inspiring directions in lung cancer diagnosis and treatment.

[1] Department of Thoracic Surgery, The First Hospital of Jilin University, 1 Xinmin Street, Changchun, Jilin 130021, PR China. [2] Division of Clinical Research, The First Hospital of Jilin University, 1 Xinmin Street, Changchun, Jilin 130021, PR China. ✉email: wmxt@jlu.edu.cn; l_w01@jlu.edu.cn

Lung cancer is a major global cause of cancer-related mortality, resulting in over one million deaths annually[1]. Based on histology, lung cancer is classified into small-cell lung cancer (SCLC) and non-small cell lung cancer, primarily consisting of lung adenocarcinoma and lung squamous cell carcinoma[2]. The pathogenesis of lung cancer is a multifaceted process involving various risk factors[3], with cancer cells' ability to maintain internal homeostasis playing a crucial role[4]. This implies that cancer cells need to regulate material turnover, particularly protein turnover, to sustain metabolic equilibrium. Thus, a high level of proteolytic system activity is indispensable for the rapid proliferation of tumor cells[5].

Cathepsins represent a group of lysosomal proteolytic enzymes that play an important role in maintaining cellular homeostasis[6]. In humans, the most well-known cathepsins belong to the papain superfamily of cysteine proteases[7]. They are integral to almost all physiological and pathophysiological cellular processes, such as protein and lipid metabolism, autophagy, antigen presentation, growth factor receptor recycling, cellular stress signaling, extracellular matrix degradation, and lysosome-mediated cell death[8]. Due to their involvement in these important processes, various cathepsins play critical roles in different diseases, including tumors[9].

Recent studies have unveiled the roles of several cathepsins, including cathepsin B[10,11], cathepsin L[12], and cathepsin S[13], in promoting or suppressing tumors in various cancers, such as breast, ovarian, pancreatic, and colorectal cancer[14,15]. However, only a limited number of observational studies and clinical trials have investigated the association between cathepsins and lung cancer. Previous studies reported elevated levels of cathepsin B and L in lung cancer patients[16]. Furthermore, findings from several studies confirmed the association of cathepsin B[17], cathepsin F[18], cathepsin H[19], and cathepsin S[20] with the survival of lung cancer patients. However, the roles of individual cathepsins can vary dramatically among different tumor subtypes[21], and the causality between various types of cathepsins and the risk of different histological lung cancers has not been adequately studied. Therefore, further investigation is necessary to elucidate the causal association between different types of cathepsins and the risk of lung cancer subtypes.

With the advancement of genomics, there is increasing evidence revealing the role of heritability in disease etiology[22]. Mendelian randomization (MR), relying on genome-wide association studies (GWAS), utilizes one or more genetic variants as instrumental variables (IVs) that are strongly associated with the exposure of interest and unaffected by confounders. MR studies can infer the causal effects of exposure on an outcome[23]. In this context, MR analyses were conducted to investigate the causal effects of different types of cathepsins on the risk of lung cancer and its histological subtypes through both univariable and multivariable MR methods.

## Results
### Defining the causal link between various cathepsins and different histological subtypes of lung cancer.
To assess the influence of various cathepsins on the risk of lung cancer subtypes, Two-Sample MR analyses involving nine cathepsins (cathepsin B, E, F, G, H, L2, O, S, and Z) and the overall risk as well as different histological subtypes of lung cancer was firstly performed. The findings of the univariable MR analysis (Fig. 1) revealed that high levels of cathepsin H increased the risk of overall lung cancer (Inverse-Variance Weighted (IVW): $p = 3.357 \times 10^{-5}$, OR = 1.060, 95% confidence interval (CI) = 1.035–1.102). This effect was consistently observed in lung adenocarcinoma (IVW: $p = 2.598 \times 10^{-4}$, OR = 1.080, 95%

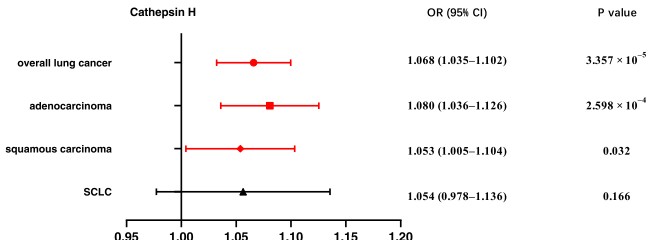

**Fig. 1 Forest plot of univariable Mendelian randomization analysis for cathepsin H and Lung cancer risk.** We conducted inverse-variance weighted analyses to evaluate the causal relationship between cathepsin H and overall lung cancer, adenocarcinoma, squamous cell carcinomas, and small-cell lung cancer. (Highlighted in red are statistically significant results, and error bars represent 95% confidence intervals).

CI = 1.036–1.126). These consistent significant associations were further corroborated by the weighted median and MR-Egger approaches (Table 1). Additionally, weaker positive effects were observed for cathepsin H levels and the risk of squamous cell carcinomas, and cathepsin G levels and the risk of adenocarcinoma, respectively, only by the IVW method ($p = 0.032$, OR = 1.053, 95% CI = 1.005–1.104; $p = 0.041$, OR = 1.095, 95% CI = 1.004–1.195) (Table 1). Furthermore, both the MR-Egger intercept and MR-PRESSO global tests provided no evidence of directional pleiotropy for any of these causal associations in Supplementary Table 1. However, the IVW method did not reveal any causal associations between the other types of cathepsins and overall lung cancer or its major histological subtypes (Table 1).

To explore the possibility of reverse causality, we conducted reverse MR analyses. These results in supplementary table 2 indicated a lack of reverse causality between cathepsin H and the risk of lung cancer and adenocarcinoma. However, the reverse MR analysis provided evidence that squamous carcinoma elevated cathepsin B levels (IVW: $p = 0.0328$, OR = 1.189, 95% CI = 1.014–1.395; and weighted median: $p = 0.038$, OR = 1.224, 95% CI = 1.011–1.481), and the $p$-values of the MR-Egger intercept and MR-PRESSO global test showing no signs of directional pleiotropy (0.826 and 0.804, respectively) (Supplementary Table 2). No evidence supported a causal association between any other histological subtypes of lung cancer and various types of cathepsins.

Moreover, we conducted multivariable MR to assess the genetic predisposition involving multiple cathepsins in relation to the risk of different histological subtypes of lung cancer. The results revealed that even after adjusting for other types of cathepsins, elevated cathepsin H levels retained a robust association with an increased risk of overall lung cancer (IVW: $p = 1.460 \times 10^{-4}$, OR = 1.070, 95% CI = 1.033–1.109) and adenocarcinoma risk (IVW: $p = 8.854 \times 10^{-5}$, OR = 1.094, 95% CI = 1.046–1.144) (Fig. 2). However, no statistically significant causal association was observed between cathepsin H and squamous cell carcinomas, or between cathepsin G and adenocarcinoma, after adjusting for other types of cathepsins, the same as the other types of cathepsins and overall lung cancer or its different histological subtypes. Moreover, horizontal pleiotropy was not indicated by the MR-Egger intercept analysis in Supplementary Table 3.

### Subgroup MR analyses stratified by smoking behavior.
Given the substantial number of lung cancer patients with a history of smoking, and the influence of smoking behavior on lung cancer development, we conducted an in-depth analysis of the causal association between various cathepsins and lung cancer risk stratified by smoking behavior (ever and never smoking). Univariable MR analysis results revealed that elevated cathepsin H

**Table 1 Causal association of cathepsins on lung cancer and its histological subtypes estimated by univariable Mendelian randomization analysis.**

| Cathepsin | SNPs | Inverse variance weighted | | MR-Egger | | Weighted median | |
|---|---|---|---|---|---|---|---|
| | | OR (95%CI) | *p*_value | OR (95%CI) | *p*_value | OR (95%CI) | *p*_value |
| **Cathepsin B** | | | | | | | |
| Overall lung cancer | 16 | 1.018 (0.977–1.060) | 0.399 | 1.025 (0.933–1.126) | 0.619 | 1.004 (0.948–1.064) | 0.884 |
| Adenocarcinoma | 16 | 1.032 (0.976–1.092) | 0.268 | 1.104 (0.970–1.256) | 0.156 | 1.043 (0.965–1.127) | 0.293 |
| Squamous carcinoma | 14 | 1.050 (0.982–1.124) | 0.155 | 1.035 (0.888–1.206) | 0.670 | 1.020 (0.933–1.114) | 0.667 |
| SCLC | 12 | 0.956 (0.821–1.114) | 0.565 | 0.781 (0.541–1.126) | 0.214 | 0.862 (0.742–1.002) | 0.053 |
| **Cathepsin E** | | | | | | | |
| Overall lung cancer | 10 | 1.036 (0.974–1.103) | 0.257 | 1.059 (0.920–1.218) | 0.449 | 1.026 (0.949–1.109) | 0.518 |
| Adenocarcinoma | 10 | 1.009 (0.926–1.099) | 0.844 | 1.097 (0.902–1.335) | 0.380 | 0.978 (0.870–1.099) | 0.707 |
| Squamous carcinoma | 9 | 0.997 (0.897–1.108) | 0.950 | 1.065 (0.786–1.443) | 0.695 | 0.975 (0.848–1.121) | 0.719 |
| SCLC | 10 | 1.096 (0.936–1.282) | 0.254 | 1.316 (0.908–1.907) | 0.185 | 1.230 (0.991–1.527) | 0.060 |
| **Cathepsin F** | | | | | | | |
| Overall lung cancer | 11 | 0.985 (0.939–1.033) | 0.540 | 1.051 (0.924–1.196) | 0.471 | 0.980 (0.919–1.044) | 0.523 |
| Adenocarcinoma | 11 | 1.004 (0.941–1.071) | 0.899 | 1.126 (0.910–1.394) | 0.302 | 1.035 (0.944–1.135) | 0.462 |
| Squamous carcinoma | 11 | 1.037 (0.962–1.118) | 0.337 | 1.061 (0.868–1.298) | 0.576 | 1.055 (0.956–1.164) | 0.291 |
| SCLC | 11 | 0.962 (0.854–1.084) | 0.525 | 1.152 (0.837–1.587) | 0.408 | 0.916 (0.779–1.077) | 0.291 |
| **Cathepsin G** | | | | | | | |
| Overall lung cancer | 11 | 1.036 (0.973–1.104) | 0.265 | 1.039 (0.893–1.209) | 0.629 | 1.049 (0.959–1.148) | 0.293 |
| Adenocarcinoma | 11 | 1.095 (1.004–1.195) | 0.041 | 1.184 (0.986–1.421) | 0.103 | 1.069 (0.950–1.204) | 0.269 |
| Squamous carcinoma | 10 | 1.039 (0.931–1.160) | 0.495 | 1.023 (0.773–1.355) | 0.877 | 1.112 (0.957–1.292) | 0.164 |
| SCLC | 11 | 1.076 (0.874–1.325) | 0.489 | 0.987 (0.631–1.543) | 0.955 | 1.031 (0.813–1.308) | 0.799 |
| **Cathepsin H** | | | | | | | |
| Overall lung cancer | 10 | 1.068 (1.035–1.102) | $3.357 \times 10^{-5}$ | 1.077 (1.017–1.140) | 0.036 | 1.074 (1.038–1.111) | $4.405 \times 10^{-5}$ |
| Adenocarcinoma | 10 | 1.080 (1.036–1.126) | $2.598 \times 10^{-4}$ | 1.093 (1.026–1.166) | 0.026 | 1.091 (1.043–1.141) | $1.674 \times 10^{-4}$ |
| Squamous carcinoma | 11 | 1.053 (1.005–1.104) | 0.032 | 1.035 (0.958–1.118) | 0.406 | 1.031 (0.977–1.088) | 0.270 |
| SCLC | 9 | 1.054 (0.978–1.136) | 0.166 | 1.020 (0.916–1.135) | 0.733 | 1.034 (0.948–1.127) | 0.448 |
| **Cathepsin L2** | | | | | | | |
| Overall lung cancer | 11 | 1.002 (0.944–1.064) | 0.942 | 1.124 (0.952–1.326) | 0.200 | 1.005 (0.924–1.093) | 0.913 |
| Adenocarcinoma | 10 | 1.024 (0.939–1.117) | 0.594 | 1.076 (0.867–1.337) | 0.524 | 1.023 (0.912–1.147) | 0.701 |
| Squamous carcinoma | 11 | 0.998 (0.878–1.136) | 0.982 | 0.985 (0.712–1.361) | 0.930 | 0.967 (0.837–1.117) | 0.651 |
| SCLC | 9 | 1.027 (0.873–1.208) | 0.750 | 1.115 (0.754–1.647) | 0.602 | 1.002 (0.814–1.235) | 0.982 |
| **Cathepsin O** | | | | | | | |
| Overall lung cancer | 11 | 0.969 (0.913–1.029) | 0.306 | 0.900 (0.793–1.022) | 0.138 | 0.987 (0.913–1.067) | 0.744 |
| Adenocarcinoma | 11 | 0.954 (0.878–1.036) | 0.261 | 0.860 (0.723–1.023) | 0.122 | 0.966 (0.872–1.071) | 0.512 |
| Squamous carcinoma | 11 | 0.978 (0.889–1.076) | 0.654 | 0.981 (0.803–1.199) | 0.857 | 1.007 (0.887–1.143) | 0.916 |
| SCLC | 8 | 1.109 (0.930–1.323) | 0.250 | 1.049 (0.656–1.678) | 0.847 | 1.098 (0.863–1.397) | 0.448 |
| **Cathepsin S** | | | | | | | |
| Overall lung cancer | 20 | 0.996 (0.953–1.041) | 0.875 | 0.933 (0.874–0.995) | 0.050 | 0.951 (0.906–0.999) | 0.046 |
| Adenocarcinoma | 21 | 1.026 (0.978–1.075) | 0.294 | 0.961 (0.885–1.043) | 0.352 | 0.998 (0.934–1.066) | 0.943 |
| Squamous carcinoma | 23 | 1.009 (0.956–1.065) | 0.734 | 0.927 (0.840–1.023) | 0.148 | 0.960 (0.889–1.037) | 0.301 |
| SCLC | 20 | 0.970 (0.889–1.058) | 0.492 | 0.884 (0.764–1.024) | 0.117 | 0.930 (0.827–1.046) | 0.224 |
| **Cathepsin Z** | | | | | | | |
| Overall lung cancer | 20 | 0.984 (0.947–1.023) | 0.410 | 0.884 (0.764–1.024) | 0.117 | 0.930 (0.824–1.049) | 0.237 |
| Adenocarcinoma | 20 | 1.000 (0.948–1.056) | 0.989 | 0.884 (0.764–1.024) | 0.117 | 0.930 (0.824–1.049) | 0.235 |
| Squamous carcinoma | 20 | 0.988 (0.929–1.052) | 0.716 | 0.884 (0.764–1.024) | 0.117 | 0.930 (0.825–1.048) | 0.232 |
| SCLC | 20 | 0.946 (0.857–1.044) | 0.269 | 0.884 (0.764–1.024) | 0.117 | 0.923 (0.825–1.048) | 0.233 |

levels significantly increased the risk of lung cancer among ever smokers ($p = 2.429 \times 10^{-6}$, OR = 1.095, 95% CI = 1.055–1.138) (Fig. 3). Similarly, no associations between the other types of cathepsins and lung cancer in individuals with a smoking history were found (Supplementary Data 1). For the never-smoker subgroup, none of the assessed associations were significant (Supplementary Data 1). In all the aforementioned analyses, no evidence of horizontal pleiotropy was detected through the MR-PRESSO global test and MR-Egger intercept ($p > 0.05$) (Supplementary Data 1). Furthermore, reverse MR analysis was performed, the results of which showed no reverse causality between cathepsin H levels and lung cancer risk among individuals with a history of smoking (Supplementary Data 1). Moreover, multivariable MR analysis confirmed a significant direct effect of cathepsin H on lung cancer risk among individuals with a smoking history ($p = 1.777 \times 10^{-5}$, OR = 1.092, 95%

CI = 1.049–1.137) (Fig. 4), with the results of the MR-Egger intercept analysis again indicating no existence of directional horizontal pleiotropy (Supplementary Data 1).

**Determining the potential mediation effects.** The abovementioned results have revealed that cathepsin H increases the risk of lung cancer among individuals with smoking behavior. Given that smoking is a well-accepted risk factor for lung cancer[24], we analyzed the possibility that cathepsin H acted as a mediator between smoking and lung cancer. Two-step MR[25] was used to explore the mediation effects of cathepsin H. The results showed no significant causal effects between smoking initiation and cathepsin H (IVW OR = 0.919, 95% CI = 0.720–1.172, $p = 0.494$), and vice versa (IVW OR = 1.002, 95% CI = 0.992–1.011, $p = 0.752$). Subsequently, Bayesian

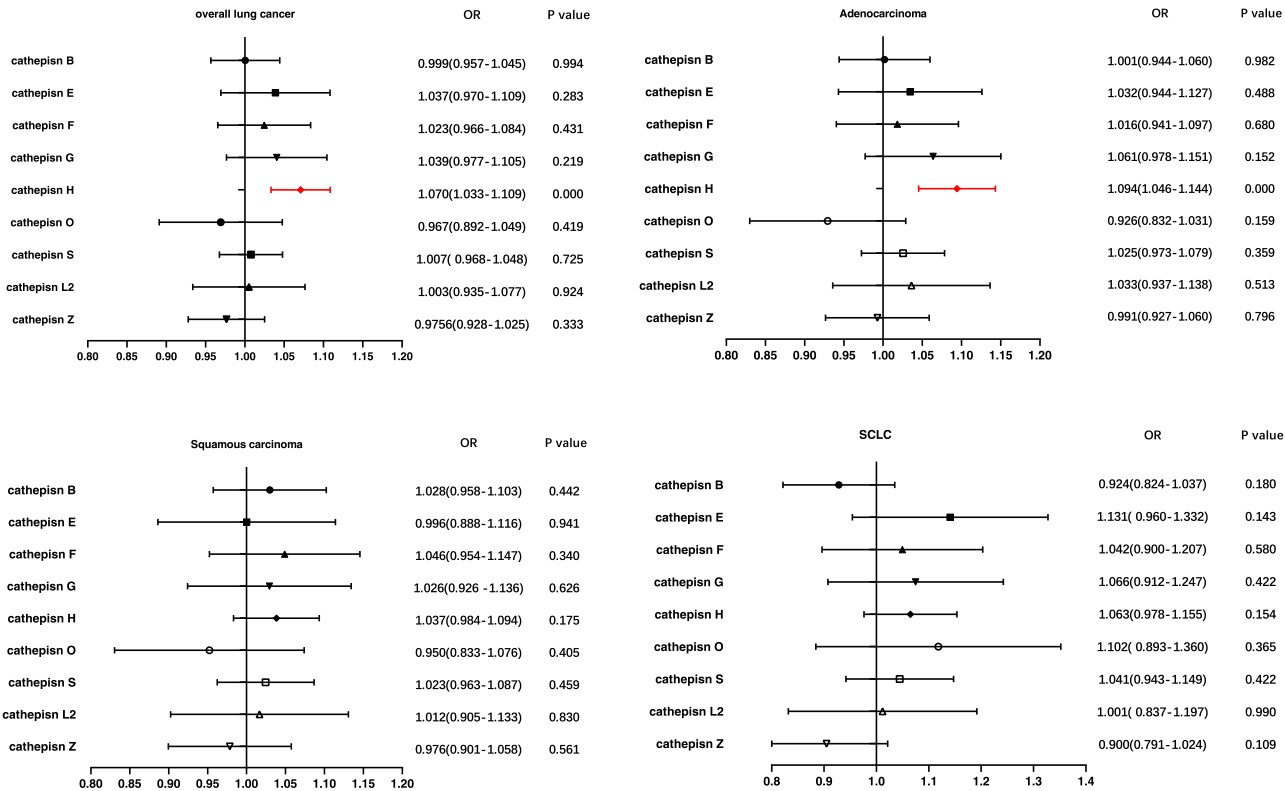

**Fig. 2 Forest plot of multivariable Mendelian randomization analysis for various cathepsins and lung cancer risk.** We employed the inverse-variance weighted method to investigate the causal associations between nine cathepsins (cathepsin B, E, F, G, H, L2, O, S, and Z) and overall lung cancer, adenocarcinoma, squamous cell carcinomas, and small-cell lung cancer (SCLC). (Highlighted in red are statistically significant results, and error bars indicate 95% confidence intervals).

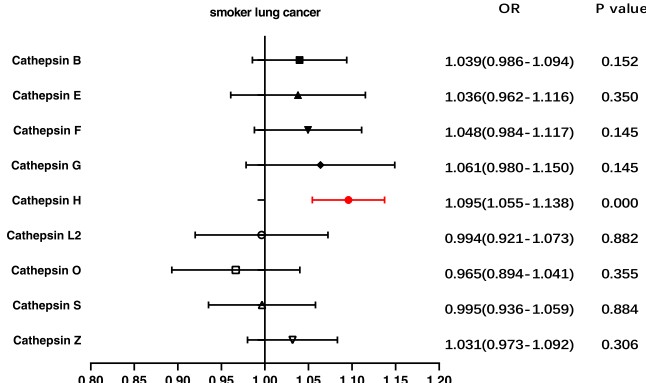

**Fig. 3 Forest plot of univariable Mendelian randomization analysis for nine cathepsins and lung cancer risk among smokers.** We utilized the inverse-variance weighted method to analyze the causal associations between different cathepsins and lung cancer in individuals with a history of smoking. (Highlighted in red are statistically significant results, and error bars represent 95% confidence intervals).

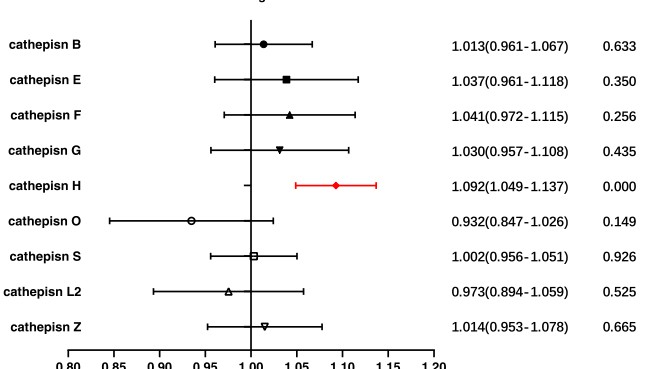

**Fig. 4 Forest plot of multivariable Mendelian randomization inverse-variance weighted analysis for nine cathepsins and lung cancer risk among smokers.** The inverse-variance weighted method was employed to investigate the causal relationships between nine cathepsins (cathepsin B, E, F, G, H, L2, O, S, and Z) and lung cancer in individuals with a history of smoking. (Highlighted in red are statistically significant results, and error bars indicate 95% confidence intervals).

colocalization analysis was performed to putatively identify whether cathepsin H drove the risk of lung cancer among ever-smokers by sharing pathway effects with smoking. *CTSH*, the gene encoding cathepsin H, is located in chromosomal region 15q25.1, and cathepsin H-related variants also reside in this region. The results of the colocalization analysis indicated no shared variants between cathepsin H and smoking for the *CTSH* locus (posterior probability = 0.004). Furthermore, we focused on smoking-related variants and performed colocalization analysis

for each candidate variant using a 10,000 kb window around the target single-nucleotide polymorphisms (SNPs). The findings revealed that all posterior probabilities for these two traits were less than 0.5. In summary, we found no valid evidence supporting a shared causal variant between the two traits. Consequently, we concluded that a high level of cathepsin H was a hazardous risk factor for developing lung cancer, rather than a mediator of the causal relationship between smoking and lung cancer.

## Discussion

The development and progression of malignant tumors involve a highly complex process in which proteolytic events play crucial roles[26]. Among the important members associated with these events, cathepsins have attracted considerable interest. In this study, we systematically analyzed the causal link between nine different cathepsins and the risk of various histological subtypes of lung cancer using genetic instruments. To our knowledge, this is the large-scale genetic consortia-based MR analysis to establish causality between cathepsins and lung cancer. By integrating findings from univariable analysis, multivariable analysis, mediation analysis, and colocalization analysis, we concluded that cathepsin H is a significant risk factor for lung cancer, especially in individuals with a history of smoking, and no reverse causality for cathepsin H was found.

The analyses conducted in this study demonstrated that cathepsin H increased the risk of overall lung cancer, adenocarcinoma, and lung cancer among smokers. The results obtained from the IVW methods were consistent with other complementary methods and did not suggest pleiotropy or reverse causality. In contrast, we found no significant association between cathepsin H and lung cancer in individuals without a history of smoking in this study. Given that the Transdisciplinary Research in Cancer of the Lung (TRICL) GWAS data for lung cancer stratified by smoking behavior included only 9859 never smokers out of 50,036, it remains unclear whether this null effect reflects the ground truth or results from inadequate statistical power. Further research is warranted.

The conclusions provided herein clarify the partial association between cathepsin H and lung cancer reported in previous observational research and clinical studies[19]. However, observational research has indicated that cathepsin H is most significantly associated with squamous cell carcinomas, an association not supported by the current MR analyses. Our results demonstrated only a weak causal link between cathepsin H levels and the risk of squamous cell carcinomas in univariable analysis using the IVW method. When other types of cathepsins were adjusted in the multivariable analysis, no statistical difference was found, possibly due to functional compensation by other family members. Multivariable MR analysis might help mitigate these potential biases that can affect conventional observational studies. Therefore, except for overall lung cancer, adenocarcinoma, and lung cancer among smokers, the current evidence is insufficient to establish any causal link between cathepsin H and squamous cell carcinomas or SCLC.

Findings from previous studies[24,27] have demonstrated that smoking behavior has a noteworthy impact on the development of lung cancer. This may introduce notable biases into the relationship between cathepsin H and the risk of lung cancer. In addition to univariable MR, both mediation MR analysis and colocalization analysis were carried out to assess potential biases introduced by smoking behavior. The findings indicated that cathepsin H has a causal effect on the risk of lung cancer, rather than serving as a mediator in the pathway from smoking to lung cancer.

Due to its unique endopeptidase activity, cathepsin H, a lysosomal cysteine protease, plays a prominent role in physiological and pathological processes[28]. Previous studies have explored possible mechanisms related to cathepsin H and tumors, suggesting that the effects of cathepsin H on tumors may be linked to its unique role in the establishment and development of tumor vasculature[29]. Additionally, cathepsin H participates in the degradation of the extracellular matrix[30] and the activation of the extracellular signal-regulated kinase[31], promoting tumor cell migration and invasion. A distinguishing feature of human lungs is the abundance of cathepsin H in the alveolar space[32],

contributing to the generation of lung surfactant involved in maintaining lung functions[33]. Therefore, the mechanism of cathepsin H in relation to lung cancer becomes more complex, and further research is needed to elucidate the role of cathepsin H in lung cancer.

Furthermore, the results of reverse MR analyses indicated that squamous cell carcinomas increase cathepsin B expression, explaining the high levels of cathepsin B detected in lung cancer patients in previous clinical studies and elucidating the unique role of squamous cell carcinomas[16,34]. Squamous cell carcinomas might regulate cathepsin B expression through transcription factors Ets1, Sp1, and Sp3[35], ultimately leading to immune resistance and tumor progression.

With increasing health awareness, tumor screening is becoming increasingly popular. Serum marker detection offers considerable advantages in tumor screening in terms of convenience and speed of detection. This study utilized MR analysis, relying on genetic variants, to explore the causal effect of various cathepsins on different subtypes of lung cancer. The integration of multivariate and reverse MR analysis minimized confounding and reverse causation bias, while mediation analysis and colocalization analysis ruled out mediation effects. These analyses yielded robust results and strengthened the final causal inference. This collective strategy can be utilized to search for and investigate effective tumor markers. However, it is important to note that the individuals included in this study are all of European descent, limiting the generalizability of the conclusions to other racial groups.

In conclusion, the primary genetic evidence from this study reveals that high levels of cathepsin H increase the risk of lung cancer, particularly adenocarcinoma and lung cancer among smokers. Additionally, squamous cell carcinoma may play an important role in regulating cathepsin B expression. This insight may aid in identifying biochemical markers for the prediction, screening, early diagnosis, and prognosis of lung cancer. Moreover, protease inhibitors targeting the specialized cathepsins associated with each histological subtype of lung cancer may offer a potential direction for effective lung cancer treatment.

## Methods

**Instrumental variables**. Genetic instruments for assessing the levels of various cathepsins (μg/L) were obtained from the INTERVAL study, which included 3301 European individuals[36]. All donors were asked to complete the trial consent, and the INTERVAL study was approved by The National Research Ethics Service (11/EE/0538). Summary data can be accessed at https://gwas.mrcieu.ac.uk. Selection of cathepsin-related IVs for MR analyses followed specific criteria: (a) an $r^2$ measure of LD among instruments $<0.001$ within a 10,000 kb window; (b) $p$-values below the genome-wide significant level identified in the corresponding study ($5 \times 10^{-6}$; this value was established in line with the limitation of the sample size). The meta-analysis of GWAS of smoking included 1,232,091 European individuals[37], with the cutoff values of independently associated SNPs established as $p < 5 \times 10^{-8}$ and $r^2 < 0.001$. The included SNPs of exposure data are detailed in Supplementary Data 2.

**Genetic association of SNPs with lung cancer risk**. Summary statistics for lung cancer risk, including log odds ratio (OR) estimates and standard errors for instrumental SNPs, were obtained from the TRICL https://www.ebi.ac.uk/gwas. These data resulted from an aggregated GWAS analysis of lung cancer, including 29,836 cases and 55,586 controls[38]. The study also provided associations between instrumental SNPs and different histological subtypes of lung cancer, including 11,273

adenocarcinomas, 7426 squamous cell carcinomas, and 2664 SCLC cases. Subgroup analyses were conducted based on smoking status, including smokers (23,223 cases and 16,964 controls) and never smokers (2355 cases and 7504 controls), limited to individuals of European descent. All participants provided informed written consent, and all studies were reviewed and approved by institutional ethics review committees at the involved institutions.

**Statistics and reproducibility**. MR utilizes genetic variants as IVs to ascertain whether an exposure causally impacts an outcome. A valid IV must meet three core criteria: First, it should be highly correlated with the exposure. Second, a SNP must not be pertinent to traits that would confound the relation between the exposure and the outcome. Lastly, certain variants cannot be associated with the outcome via other paths rather than the exposure. A SNP is considered to have horizontal pleiotropy when the last two assumptions are violated.

In this MR study, the IVW was employed as the primary method to estimate an overall effect size[39]. Briefly, the influence of each variant on the risk of the disease under investigation was weighted by its effect on the exposure using the Wald ratio method in IVW. Subsequently, these individual MR estimates were amalgamated to attain an overall summary value employing a random-effect inverse variance meta-analysis. Complementary methods, including MR-Egger[40] and weighted median[41], were used to validate the robustness of the MR results. Briefly, MR-Egger regression[40] is a weighted linear regression of the SNP-outcome association on the SNP-exposure associations, and the estimator of the weighted median method[41] is a median in which individual MR estimates are weighted proportionally to their precisions, as its name implied. MR analyses (including IVW, MR-Egger, and weighted median) were executed using the R TwoSampleMR package[42].

Various sensitivity analyses and statistical tests were conducted to evaluate the validity of assumptions. Cochran's Q test was used to estimate the heterogeneity of the SNPs. A $p$-value > 0.05 indicated a lack of heterogeneity. The random effects model was applied when significant heterogeneity among the SNPs existed; otherwise, a fixed effects model was used[43]. MR-PRESSO global test and MR-Egger intercept were employed to identify outliers and horizontal pleiotropic effects[44]. The intercept of MR-Egger represents the average pleiotropic effect (intercept $p$ value < 0.05) and the slope could produce a robust pleiotropy MR estimate. The MR-PRESSO outlier test was used to correct for horizontal pleiotropy by removing or down-weighting the outliers when the horizontal pleiotropy was significant ($p$-value of MR-PRESSO global test < 0.05). Additionally, the MR-PRESSO distortion test was used to identify significant distortion in causal estimates before and after removing outliers. MR-PRESSO global, outlier, and distortion tests were performed using the R MR-PRESSO package[44]. Leave-one-out analysis was also conducted to identify SNPs with potential extreme influence on estimates and further evaluate the reliability of the results.

Multivariable MR, an extension of standard univariable MR, was used to consider multiple cathepsins when analyzing their causal effects on different lung cancer subtypes and estimating the direct causal effects of each exposure in a single analysis, employing the "MendelianRandomization" package[43]. Reverse MR analyses, treating lung cancer as the exposure and cathepsins as the outcomes, were performed to evaluate reverse causality and justify the existence of bidirectional causality. In these reverse MR analyses, the same GWAS datasets as the above mentioned were used, the IVs for lung cancer were selected from TRICL, and the abundance levels of cathepsins from the INTERVAL study were

used as outcomes. Two-step MR[25], a sequence of two MR analyses connected by a shared variable, was employed in mediation analysis to assess whether one trait acts as a mediator, such as whether the cathepsins family lies in between the path from smoking behavior to lung cancer.

Colocalization analysis was conducted using the Coloc package[45] to test whether common genetic variants within a given region were shared between two traits. In brief, Bayesian approach calculated a posterior probability of two traits sharing common genetic variants within the same genomic region. All statistical analyses were performed using R software version 4.1.1.

**Reporting summary**. Further information on research design is available in the Nature Portfolio Reporting Summary linked to this article.

## Data availability
The raw data analyzed during the current study were available in public databases https://gwas.mrcieu.ac.uk. and https://www.ebi.ac.uk/gwas. The detailed accession number of involved datasets and summary data (including specific IVs) of the main results, along with source data underlying Figs. 1–4, are available in Supplementary Data 3.

## Code availability
All packages for data analysis used in this study were open source in R software (version 4.1.1; R Development Core Team).

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

## Acknowledgements
We thank Sagesci (www.sagesci.cn) for its linguistic assistance during the preparation of this manuscript. This work was supported by Jilin Provincial Science and Technology Development Plan Project (20220204115YY); Jilin Provincial Science and Technology Development Plan Project, Natural Science Foundation of Jilin Province (YDZJ202201-ZYTS121 and YDZJ202301ZYTS007); and Jilin Provincial Science and Technology Research projects of Education Office (JJKH20231215KJ).

## Author contributions
S.T. and W.L. conceived and designed the experiment; J.L. ran the analysis and verified the underlying data; J.L. and S.T. wrote the original manuscript. M.T., W.L. and X.G. involved in data interpretation. All authors have read and approved the final version of the manuscript.

## Competing interests
The authors declare no competing interests.
