## [Peer review file · Communications Biology]

Reviewer #1 (Remarks to the Author):

The authors report results from Mendelian randomization analyses of cathepsins and lung cancer and subtypes of European ancestry. The authors identified a causal relationship between high level of cathepsin H and increased risk of overall lung cancer using a univariable and multivariable MR methods. Reverse casualties, pleiotropy and colocalization were carefully checked using proper methods. As far as I could evaluate the statistical analyses are well-performed and I have no major comments or concerns regarding those. The methods are appropriate, but the manuscript is not well written in the sense that the results are not clearly presented, the tables are missing and not properly cited, and the descriptions are vague and not complete. I have the following suggestions for improvement.

- 1) Line 63. Please cite the table or figure where you first present the numbers.
- 2) Line 66. Please present the results from the weighted median and MR-Egger approaches in supplemental table(s) and also cite them.
- 3) Line 70. Please present the results of MR-egger and Mr-PRESSO tests as a supplemental table(s).
- 4) Line 80. Please cite the table of the reverse causality results.
- 5) Line 81. "aforementioned positive results", do you mean "significant results?"
- 6) Line 88. Missing "higher" in front of "cathepsin H still ..."
- 7) Line 89. Please cite the figure where you first present the numbers.
- 8) Line 91. What is "CTSH"?
- 9) Line 105. This sentence does not make sense. When have the authors ever investigated association between cathepsins and the tobacco-smoking status?
- 10) Line 109. This saying is confusing. Odds ratio is always positive (>0).
- 11) Line 114. Please indicate which univariable MR method.
- 12) Line 146. Please explain why the authors particularly looked at smokers. We all know smoking is a risk factor for lung cancer. If we can find a causal factor for lung cancer among non-smokers, it will be more scientifically interesting.
- 13) Line 161. Please explain: how does the current study suggest that smoking behavior is a confounder? I couldn't find any analysis and discussion regarding this statement.
- 14) Line 208. Please cite the lung cancer GWAS by "(TRICL)".

Reviewer #2 (Remarks to the Author):

Summary:

This study mainly explored the causal association between cathepsins protein family and lung cancer risk through Mendelian randomization, mediation analysis and colocalization methods, and first verified the causal association between cathepsin H and total lung cancer risk and lung adenocarcinoma risk by using MR method. Mendelian mediation analysis and colocalization analysis further showed that the effect of cathepsin H on lung cancer risk is independent of smoking status. While several previous study has reported the association between cathepsins and lung cancer risk, this study elucidate the causal association between different types of cathepsin and the risk of lung cancer subtypes. However, the implication of this study in clinical practice is unclear. There are still some problems in this paper, mainly in the of results and methods, as detailed below.

Comments:

- Line 21: The article stated that there is no reverse causality, but it also reported that lung squamous cell carcinoma increased the level of cathepsin B due to reverse causality.
- Line 60: The article uses several related terms including bidirectional MR, univariable MR, multivariable MR, mediation MR, etc., but none of them were introduced in the methods section, this part should be clarified.
- Line 66: The article uses methods including IVW method, MR-Egger, weighted median method, MR-PRESSO global test and other methods for the quality control and robustness analysis of instrumental variables in mendelian randomization analysis, but they should be briefly introduced in the method section. This part should be added to the text.
- Line 70: Sensitivity analysis results in MR analysis, such as MR-egger analysis results are not

shown in the article.

Line 70: The article uses multiple cathepsin-related SNPs for mendelian randomization analysis, but didn't report the summary statistic information of these them, such as Chr, Pos, Beta, SE, A1_freq, etc.

Line 73: Instrumental variables used in MR analyzes should be reported their units.

Line 81: The results of the reverse mendelian randomization analysis in the article report that there is a causal effect between squamous cell carcinoma and cathepsin B, but the P values of the IVW method and the weighted median method are marginal, the author should report the results of the MR-Egger test to demonstrate its reliability.

Methodology

Line 200:

1. In MR analysis, the instrumental variables always are strong associated with exposure variable. The author should clarify why the instrumental variable P value threshold of the cathepsin family was set at $5e-06$, while the smoking-related instrumental variable threshold was set at $5e-08$ in the methods section.

2. In the reverse MR analysis, what is the instruments for lung cancer? What is the data of outcome GWAS?

3. Line 227: MR-PRESSO global test and MR-egger intercept test can be used to assess whether there is horizontal pleiotropy and the MR analysis results after correcting horizontal pleiotropy, but they do not indicate outliers; The application purpose and method of each part should be indicated here.

4. The two ways of writing "MR-Egger intercept test" or "MR-egger intercept test" should be unified
Line 146: Mediation analysis and colocalization analysis only seem to indicate that there is no common genetic basis between the genetic determinants of cathepsin H and smoking, but this cannot directly prove that the risk of cathepsin H for lung cancer is completely independent of other underlying factors.

Line 163: Please explain what is meant by "removing confounding factors caused by smoking by comparing univariate MR, mediator MR and colocalization analysis"? How to identify the confounding factors?

Minor:

Line 96: Figure 2, The "OR" in the picture should be changed to "OR (95%CI)", and annotations below images of multivariable MR analysis should include adjusted covariate information.

Reviewer 1

We express our great gratitude for both reviewers' invaluable suggestions. Below, we provide detailed responses addressing those points one by one.

Reviewer 1 point 1. Please cite the table or figure where you first present the numbers.

Response: We acknowledge your guidance, and in the revised version, we have made certain that all tables, figures, and supplemental tables are duly cited upon their initial appearance.

Reviewer 1 point 2. Please present the results from the weighted median and MR-Egger approaches in supplemental table(s) and also cite them.

Response: We have added the outcomes of the Weighted Median and MR-Egger methodologies into Table 1, as indicated in line 66.

Reviewer 1 point 3. Please present the results of MR-egger and Mr-PRESSO tests as a supplemental.

Response: Table S1 has been cited, containing the results of the MR-Egger and MR-PRESSO tests, as indicated in line 70.

Reviewer 1 point 4. Line 80. Please cite the table of the reverse causality results.

Response: The findings regarding reverse causality have been incorporated into Table S2, as indicated in line 83.

Reviewer 1 point 5: Line 81. "aforementioned positive results", do you mean "significant results?"

Response: The phrase "The aforementioned positive results" has been revised to indicate statistically significant outcomes between cathepsin H and the risk of lung cancer and adenocarcinoma. Corresponding sentences have been adjusted accordingly in the whole text.

Reviewer 1 point 6: Line 88. Missing "higher" in front of "cathepsin H still ..."

Response: In line 91, we have introduced the term "higher" preceding "cathepsin H still...".

Reviewer 1 point 7: Line 89. Please cite the figure where you first present the numbers.

Response: A citation for the specific figure has been included as requested.

Reviewer 1 point 8: Line 91. What is "CTSH"?

Response: The abbreviation "CTSH" has been replaced with the full term "cathepsin H" upon its first appearance in line 94.

Reviewer 1 points 9: Line 105. This sentence does not make sense. When have the authors ever investigated association between cathepsins and the tobacco-smoking status?

Response: Your constructive suggestion is highly appreciated. We concur with your assessment of the sentence's inaccuracy. In this context, our investigation revolves around the correlation between cathepsins and lung cancer stratified by smoking behavior. Specifically, the GWAS data of TRICL encompasses a subgroup of lung cancer cases categorized by smoking behavior. Our analysis, detailed under the section "Subgroup MR analyses stratified by smoking behavior," involves a systematic examination of the causal association between nine cathepsins and lung cancer patients stratified by smoking status. We performed MR analyses separately for smokers and non-smokers. The comprehensive results are provided in Table S4. Notably, cathepsin H exhibited an association with lung cancer in patients with a smoking history. We have accordingly rephrased the relevant sections for clarity in lines 105-112.

Reviewer 1 point 10: Line 109. This saying is confusing. The odds ratio is always positive (>0).

Response: Acknowledging the issue with the term "positive" ("significant" should be a better word), we have rectified this inconsistency within the main text.

Reviewer 1 point 11: Line 114. Please indicate which univariable MR method.

Response: Apologies for the oversight. We have augmented Figure 3 to show the outcomes of the univariable IVW MR analysis pertaining to the correlation between nine cathepsins and the risk of lung cancer in smokers. This information is now duly integrated into the main text.

Reviewer 1 points 12: Please explain why the authors particularly looked at smokers. We all know smoking is a risk factor for lung cancer. If we can find a causal factor for lung cancer among non-smokers, it will be more scientifically interesting.

Response: We appreciate your insightful suggestion. Given the prominent role of smoking as a well-established risk factor for lung cancer, we undertook subgroup MR analyses, stratified by smoking behavior. Our comprehensive analysis encompasses both univariable and multivariable MR approaches. We meticulously evaluated the causal association between cathepsins and lung cancer patients with smoking history using both univariable and multivariable MR analyses, the detailed results are shown in Table S4.1. We observed an association between cathepsin H and lung cancer in individuals with a history of smoking. Additionally, we examined the causal relationship between the cathepsin family and lung cancer in individuals who never smoked; however, no significant results were obtained (as indicated in Table S4.1 in lines 111-112).

Subsequently, we delved into this discrepancy in detail. We deduced that the lack of a significant association between cathepsin H and lung cancer in individuals without a smoking history within this study could potentially stem from the fact that the TRICL GWAS dataset comprised only 9,859 never smokers out of a total of 50,036 individuals. Given this, we cannot definitively determine whether this absence of effect truly reflects the actual situation or it results from insufficient statistical power. We have incorporated this explanation into the Discussion section in lines 158-162.

Conversely, considering smoking's well-established status as a significant risk factor for lung cancer, we aim to investigate whether cathepsin H acts as a mediator in the pathway connecting smoking and lung cancer. Therefore, we also conducted mediation and colocalization analyses to justify cathepsin H is indeed causally associated with lung cancer.

Reviewer 1 point 13: Line 161. Please explain: how does the current study suggest that smoking behavior is a confounder? I couldn't find any analysis or discussion regarding this statement.

Response: We realized that "confounder" is not a correct word here. Since smoking significantly affected the development of lung cancer^{1,2}, it is possible that smoking may introduce nonmarginal biases to the association between cathepsin H and the risk of lung cancer. Consequently, we investigated the impact of smoking on the causal association between cathepsin H and lung cancer by employing mediation MR and colocalization analyses. We have rephrased these sentences accordingly (lines 173-178).

Reviewer 1 point 14: Line 208. Please cite the lung cancer GWAS by "(TRICL)".

Response: We have appropriately referenced the lung cancer GWAS study conducted by TRICL in line 222.

Reviewer 2

Reviewer 2 point 1: Line 21: The article stated that there is no reverse causality, but it also

reported that lung squamous cell carcinoma increased the level of cathepsin B due to reverse causality.

Response: Our acknowledgment of this inaccuracy is noted. Our study's findings reveal that cathepsin H elevates the risk of overall lung cancer, adenocarcinoma, and lung cancer in individuals with smoking history. The reverse MR analysis indicates the absence of significant results between cathepsin H and the overall risk of lung cancer, adenocarcinoma, or lung cancer with smoking behavior. Hence, the statement in line 83, "the absence of any reverse causality in the aforementioned significant results," pertains to cathepsin H's outcomes. However, the use of lung cancer GWAS as instrumental variables and cathepsins GWAS as outcomes did yield a significant finding between lung squamous cell carcinoma and cathepsin B, suggesting that squamous carcinoma may lead to elevated cathepsin B. In brief, all results of the two-sample MR did not exhibit a bidirectional causal relationship. We realized that the two adjacent sentences, "Squamous carcinoma might lead to high-level cathepsin B" and "No reverse causality was identified through reverse MR analyses," were causing confusion in the Abstract. Therefore, in the revised version, we have removed the sentence "No reverse causality was found by reverse MR analyses" from line 21. Meanwhile, we have accurately clarified in the main text that "No reverse causality was found for cathepsin H" in line 153.

Reviewer 2 point 2. Line 60: The article uses several related terms including bidirectional MR, univariable MR, multivariable MR, mediation MR, etc., but none of them were introduced in the methods section, this part should be clarified.

Response: We have taken steps to enhance the introduction of our methodologies. This includes the clarification of bidirectional MR (lines 263-265), univariable MR (lines 236-245), multivariable MR (lines 261-263), and mediation MR (lines 267-270) approaches in the "Mendelian randomization" section of the Methods.

Reviewer 2 point 3. Line 66: The article uses methods including the IVW method, MR-Egger, weighted median method, MR-PRESSO global test, and other methods for the quality control and robustness analysis of instrumental variables in Mendelian randomization analysis, but they should be briefly introduced in the method section. This part should be added to the text.

Response: The introduction of methods encompassing the IVW method (lines 236-240), MR-Egger, weighted median method (lines 240-245), MR-PRESSO global test, and MR-Egger intercept (lines 252-259) has been incorporated into the "Mendelian randomization" section of the Methods.

Reviewer 2 point 4. Sensitivity analysis results in MR analysis, such as MR-egger analysis results are not shown in the article.

Response: We have duly cited the supplemental tables that provide insights into sensitivity analysis outcomes, including MR-Egger tests, MR-PRESSO tests, and reverse causality. These citations are marked with red text in lines 70, 83, 97, and 113 with the corresponding tables available in the supplementary materials.

Reviewer 2 point 5. Line 70: The article uses multiple cathepsin-related SNPs for Mendelian randomization analysis, but didn't report the summary statistic information of these them, such as Chr, Pos, Beta, SE, A1_freq, etc.

Response: Information regarding SNPs linked to cathepsins has been summarized within Supplement Table 5 and cited in line 218.

Reviewer 2 point 6. Instrumental variables used in MR analyses should be reported in their units.

Response: The unit " $\mu\text{g/L}$ " has been included in line 210 to denote the measurement of cathepsins levels, as per the original research.

Reviewer 2 point 7. Line 81: The results of the reverse Mendelian randomization analysis in the article report that there is a causal effect between squamous cell carcinoma and

cathepsin B, but the P values of the IVW method and the weighted median method are marginal, the author should report the results of the MR-Egger test to demonstrate its reliability.

Response: Detailed outcomes stemming from reverse MR analysis have been documented in Table S1 including the related sensitive analysis of p values of the MR-Egger intercept and MR-PRESSO global test. These showed no evidence of directional pleiotropy, and the corresponding values were 0.826 and 0.804, respectively. Corresponding references to this supplementary table have been added in the main text, specifically in lines 85–87.

Reviewer 2 point 8. Line 200: In MR analysis, the instrumental variables always are strong associated with exposure variable. The author should clarify why the instrumental variable P value threshold of the cathepsin family was set at 5e-06, while the smoking-related instrumental variable threshold was set at 5e-08 in the methods section.

Response: The summary data for the cathepsin family was derived from the Genomic Atlas of the human plasma proteome, encompassing 3,301 individuals of European ancestry. Considering the limited sample size, the genetic significance level for the cathepsin family was loosely defined as 5e-06. This rationale has been appended to lines 214-215. Conversely, for GWAS data on smoking, which encompassed 1,232,091 individuals of European descent, the default threshold of 5e-08 was applied.

Reviewer 2 point 9. In the reverse MR analysis, what is the instruments for lung cancer? What is the data of outcome GWAS?

Response: In reverse MR analysis, consistent GWAS datasets were utilized, in alignment with the aforementioned methodologies. The instrumental variable for lung cancer originated from TRICL and was selected based on the criteria of P value < 5e-08 and $r^2 < 0.001$. The outcomes involved the cathepsins family from the INTERVAL study. This clarification has been provided in lines 265-267.

Reviewer 2 point 10. Line 227: MR-PRESSO global test and MR-egger intercept test can be used to assess whether there is horizontal pleiotropy and the MR analysis results after correcting horizontal pleiotropy, but they do not indicate outliers; The application purpose and method of each part should be indicated here.

Response: The purpose and method of the MR-PRESSO global test, MR-Egger intercept test, and the outliers test have been expounded upon within the "Mendelian randomization" section of the Methods (lines 241-242, and 252-259).

Reviewer 2 point 11. The two ways of writing "MR-Egger intercept test" or "MR-egger intercept test" should be unified.

Response: We appreciate your attention to these details. The term "MR-Egger intercept test" has been uniformly employed throughout the manuscript.

Reviewer 2 point 12. Line 146: Mediation analysis and colocalization analysis only seem to indicate that there is no common genetic basis between the genetic determinants of cathepsin H and smoking, but this cannot directly prove that the risk of cathepsin H for lung cancer is completely independent of other underlying factors.

Response: Our gratitude for your insightful recommendation. We acknowledge the imprecision of the statement in question. In line with your advice, we have rephrased the sentence to assert that "a high level of cathepsin H may pose a hazardous risk for lung cancer, rather than serving as a mediator within the causal pathway linking smoking and lung cancer" in lines 141-143.

Reviewer 2 point 13. Line 163: Please explain what is meant by "removing confounding factors caused by smoking by comparing univariate MR, mediator MR, and colocalization analysis"? How to identify the confounding factors?

Response: We acknowledge the concerns raised by both reviewers regarding the term "confounder." Since smoking is a well-established lung cancer risk factor, we acknowledge

the potential for biases introduced by smoking into the association between cathepsin H and lung cancer risk. Thus, we conducted mediation MR and colocalization analyses to assess the potential biases stemming from smoking. The resultant text adjustments reflect this clarification (lines 173 - 178).

Reviewer 2 point 14. Line 96: Figure 2, The "OR" in the picture should be changed to "OR (95%CI)", and annotations below images of multivariable MR analysis should include adjusted covariate information.

Response: In Figure 2, all instances of "OR" have been replaced with "OR (95%CI)," and the legend now includes information about adjusted covariates in multivariable MR, where nine types of cathepsin family serve as covariates.

REVIEWERS' COMMENTS:

Reviewer #1 (Remarks to the Author):

The authors addressed all my concerns reassuringly. In the meantime, I suggest that the authors get editing help from someone with full professional proficiency in English to make the paper clearer.

Reviewer #2 (Remarks to the Author):

None

REVIEWERS' COMMENTS:

Reviewer #1 (Remarks to the Author):

The authors addressed all my concerns reassuringly. In the meantime, I suggest that the authors get editing help from someone with full professional proficiency in English to make the paper clearer.

Response: We have SageSCI edit the main text for English language and grammar, and the certificate of editing is attached as below.

September 14, 2023

EE-230526-M10
Total words edited: 3157

CERTIFICATE OF EDITING

I hereby certify that the attached manuscript has been meticulously edited for proper English language usage, grammar, punctuation, spelling, and overall style either by myself or under my supervision.

Manuscript Title

Mendelian randomization analyses explore the relationship between cathepsins and lung cancer

Feroz Khan, MS, PhD,
Ex. Faculty: University of California (UCSF), San Francisco, CA

Digitally
signed by
Feroz Khan
Date:
2023.09.14
14:35:00
+05'30'

Disclaimer: The authors received an editable copy of the text and were free to alter/edit/reject/accept changes after editing. Hence, the version under consideration for publication may be different from the one we have provided; in such a case, the text attached herewith should be considered authentic or unaltered. Additionally, authors may receive selective editing that excludes headings such as materials, methods, tables, figure legends, and/or bibliography. In such cases, please consider only the text attached herewith as the edited version.

Please verify the digital signature of this certificate before accepting.

Please verify the digital signature of this certificate before accepting it.

1883, 20th Avenue. San Francisco, CA 94122, USA.

Reviewer #2 (Remarks to the Author):

None

Response: Thank you very much for your decision and suggestions on our manuscript.